# Permittivity and Dielectric Loss Balance of PVDF/K_1.6_Fe_1.6_Ti_6.4_O_16_/MWCNT Three-Phase Composites

**DOI:** 10.3390/polym14214609

**Published:** 2022-10-30

**Authors:** Alexey Tsyganov, Maria Vikulova, Denis Artyukhov, Alexey Bainyashev, Vladimir Goffman, Alexander Gorokhovsky, Elena Boychenko, Igor Burmistrov, Nikolay Gorshkov

**Affiliations:** 1Department of Chemistry and Technology of Materials, Yuri Gagarin State Technical University of Saratov, 77 Polytecnicheskaya Street, 410054 Saratov, Russia; 2Department of Power and Electrical Engineering, Yuri Gagarin State Technical University of Saratov, 77 Polytecnicheskaya Street, 410054 Saratov, Russia; 3Engineering Center, Plekhanov Russian University of Economics, 36 Stremyanny Lane, 117997 Moscow, Russia

**Keywords:** three-phase composites, polyvinylidene difluoride, hollandite, carbon nanotubes, dielectric properties

## Abstract

New three-phase composites, destined for application as dielectrics in the manufacturing of passive elements of flexible electronics, and based on polymer (PVDF) matrix filled with powdered ceramics of the hollandite-like (KFTO(H)) structure (5.0; 7.5; 15; 30 vol.%) and carbon (MWCNT) additive (0.5; 1.0; 1.5 wt.% regarding the KFTO(H) amount), were obtained and studied by XRD, FTIR and SEM methods. Chemical composition and stoichiometric formula of the ceramic material synthesized by the sol–gel method were confirmed with the XRF analysis data. The influence of the ceramic and carbon fillers on the electrical properties of the obtained composites was investigated using impedance spectroscopy. The optimal combination of permittivity and dielectric loss values at 1 kHz (77.6 and 0.104, respectively) was found for the compositions containing K_1.6_Fe_1.6_Ti_6.4_O_16_ (30 vol.%) and MWCNTs (1.0 wt.% regarding the amount of ceramic filler).

## 1. Introduction

Nowadays, the composites based on different polymer matrices and characterized with enhanced dielectric properties (high permittivity and low dielectric loss tangent) attract a lot of attention for their application as flexible materials for electronic components, namely sensors and converters, energy storage elements and implantable devices [1,2,3,4,5]. Polyvinylidene difluoride (PVDF) is a prospective matrix material among other polymers due to relatively high permittivity (ε′ ≈ 10–11) [6]. PVDF (chemical formula of (C_2_H_2_F_2_)n) is a semi-crystalline polymer represented by four phases (α, β, δ and γ). Only β-phase can exhibit spontaneous polarization [7]. Although PVDF has a high permittivity relative to other polymer matrices, it is low from a general point of view and limits its potential application in electronics. That is why ferroelectric ceramic fillers are introduced into the composition of the polymer matrix to obtain composite materials characterized by high flexibility and permittivity [8]. The ceramic filler can effectively increase the composite permittivity. The materials with high permittivity, for example, BaTiO_3_, CaCu_3_Ti_4_O_12_ etc., are usually used as ceramic fillers [9,10,11,12]. The volume fraction of the ceramic filler in the polymer matrix is of about 50 vol.% to obtain high permittivity values. However, such contents of the filler can lead to reduced mechanical properties and do not allow composites to have permittivity of more than 100 and dielectric loss tangent of less than 0.05. Since the permittivity and dielectric loss tangent do not reach the desired values, and a further increase in the volume fraction of the ceramic filler leads to degradation in mechanical characteristics, the use of three-phase composites is one of the possible solutions to improve its properties. Metal nanoparticles [13,14,15] and various modifications of nanocarbon materials [16,17] are usually used as a third phase. Carbon nanotubes are attracting attention as fillers in polymer matrix composites. MWCNTs can contribute to high permittivity of composites at low content due to the effect of percolation [18,19,20]. In addition, compared to other conductive particles, MWCNTs have a larger aspect ratio and higher conductivity, which can lead to higher permittivity of composites [21]. However, along with high permittivity, high conductivity leads to high dielectric losses, making them unsuitable for electronic devices. The advantages of a three-phase composite are the possibility of increasing the permittivity and reducing the dielectric losses at low concentrations of the filler [22,23]. The dielectric properties of three-phase composites depend on types of filler and polymer matrix, including the structure, filler loading, and compatibility of the filler and polymer matrix [24]. In any case, three-phase composites have ample opportunities for regulating and optimizing the permittivity and dielectric losses.

In addition to traditional ceramic ferroelectrics, the hollandite-like potassium titanates with K_x_(Me,Ti)_8_O_16_ can be used as functional fillers [25,26]. In recent years, it has been noted that ceramics based on potassium titanates can exhibit a permittivity comparable to well-known high-K materials [27,28]. In recent publications, an increase in the permittivity of composites based on various polymer matrices filled with hollandite-like ceramic fillers is noted [29,30], and the potassium titanates doped with iron ions show the highest values of permittivity [31].

The selection of the experimental conditions to obtain polymer matrix composites, based on PVDF and providing the optimal combination of dielectric properties, namely the maximal possible permittivity and minimal possible dielectric losses, is the main issue to be resolved in this study. The main approach used to solve this problem was the determination of the upper limit of the ceramic filler contents in the presence of the MWCNT additive. This characteristic is also important to produce a composite with acceptable mechanical properties. The optimal combination of the abovementioned filler contents supporting functional properties of the investigated composite materials is the main practical task of this research.

The aim of this study is to develop a high-K polymer matrix composite characterized by low dielectric losses. A three-phase approach was adopted for these purposes using PVDF as the polymer matrix as well as K_1.6_Fe_1.6_Ti_6.4_O_16_ (KFTO(H)) Fe-doped hollandite-like potassium titanate, obtained by the sol–gel method, and MWCNTs were used as functional fillers.

## 2. Materials and Methods

### 2.1. KFTO(H) Synthesis

Fe-doped potassium titanate with hollandite-like structure (KFTO(H)) was synthesized by the sol–gel Pechini method [32,33] using analytical grade reagents without any additional purification, namely C_2_H_6_O_2_ (98.5%, Russian Standard 19710-2019, Aricon, Moskow, Russia), C_6_H8O_7_ (99.5%, Russian Standard 3652-69, Aricon, Moskow, Russia), KNO_3_ (98%, Russian Standard 4217-77, Buyskiy himicheskiy zavod, Buy, Russia), Fe(NO_3_)_3_ 9H_2_O (98%, TC 6-09-02-553-96, Buyskiy himicheskiy zavod, Buy, Russia), C_16_H_36_O_4_Ti (99%, Acros Organics, Geel, Belgium), HNO_3_ (65%, Russian Standard 4461-77, Buyskiy himicheskiy zavod, Buy, Russia), NH_4_OH (25%, Russian Standard 3760-79, Moskow, Russia). Potassium and iron nitrates were taken in a stoichiometric amount according to the chemical formula of the compound K_1.6_Fe_1.6_Ti_6.4_O_16_. To obtain a single-phase highly dispersed KFTO(H) nanopowder, the ratios of the cross-linking agent and the chelating agent of metal ions, as well as the acid residue, were varied in the following ranges Ti: C_6_H_8_O_7_ = 0.5–2; Ti: C_2_H_6_O_2_ = 2–8; Ti: NO_3_ = 1–3. The choice of the optimal ratios was based on the occurrence of a self-igniting reaction that contributes to the production of a finely dispersed powder, as well as on the phase composition of the reaction product. The experimentally determined optimal stoichiometric ratios were Ti:C_6_H_8_O_7_ = 1.5; Ti:C_2_H_6_O_2_ = 5.5; Ti:NO_3_ = 1.8, respectively. The sol was formed in the solution of all reagents at pH = 8.0. The gel was obtained by drying at 240 °C. The final calcination of the intermediate product to obtain the hollandite-like crystalline structure was carried out at 900 °C for 4 h [34].

### 2.2. Characteristics Methods

The ARL X’TRA device (Thermo Scientific, Ecublens, Switzerland) using Cu Kα radiation (λ = 0.15412 nm) was used to record the diffractogram of KFTO(H). Morphology of hollandite and carbon material, as well as the distribution of fillers and structural features of polymer matrix composites, was investigated using an ASPEX Explorer scanning electron microscope (ASPEX, Framingham, MA, USA). Fourier-transform infrared spectroscopy (FTIR) was carried out using an FT-801 FTIR spectrometer (Simex, Novosibirsk, Russia). The elemental composition of the KFTO(H) powder was determined by a BRA-135F spectrometer (BOUREVESTNIK, Saint Petersburg, Russia).

The dielectric characteristics of the composite materials were investigated using a Novocontrol Alpha AN Impedance Analyzer (Novocontrol Technologies GmbH & Co. KG, Montabaur, Germany). The measurements were carried out in the frequency range from 100 Hz to 1 MHz at room temperature with voltage amplitude of 100 mV. The permittivity (ε′, ε″ and ε), conductivity (σ), and dielectric losses (*tanδ*) were found from the experimental values of the real and imaginary parts of the impedance (Z′ and Z″) using the standard computing operations.

Both sides of the discs, produced using different composite materials, were covered by silver-containing adhesive (trademark Contactol K13, OOO Adecvat, Moscow, Russia) and dried at 120 °C to obtain the electrodes for use in the impedance measurements.

### 2.3. Composite Production

The following raw materials were used to obtain KFTO(H)/MWCNT/PVDF composites: granulated poly(vinylidene fluoride) (PVDF, Sigma Aldrich, Mw~530,000, St. Louis, MO, USA), N, N—Dimethylformamide (DMF, C_3_H_7_NO, analytical grade, Russian Standard 20289-74, Ekos-1, Moscow, Russia), “Taunit-M” multi-walled carbon nanotubes (MWCNTs, NanoTechCenter, Tambov, Russia).

The preparation of composites was carried out according to the methodology presented in Figure 1. At the first stage, the mixtures of KFTO(H) and MWCNT powders were prepared with varying concentrations of MWCNTs (0, 0.5 wt.%, 1.0 wt.%, 1.5 wt.%). Homogenization and grinding of powders were carried out in a vibrating Fritsch Pulverisette 0 micro-mill for 1 h. Then, the mixtures of powders were introduced into the PVDF polymer matrix at concentrations of 5.0, 7.5, 15, 30 vol.% (using the theoretical densities of KFTO(H) (3.7 g/cm^3^) and PVDF (1.11 g/cm^3^)) according to the following procedure: PVDF granules were dissolved in DMF by stirring with a mass ratio of PVDF:DMF = 5:95, then the KFTO(H)/MWCNT powdered mixtures were added to the solution and stirred at 60 °C for 2 h by a magnetic stirrer with periodic ultrasonic treatment to ensure the best homogenization. The dispersions were then poured onto aluminum plate substrates and dried at 120 °C. The resulting composites were subjected to hot pressing at a temperature of 160 °C under a pressure of 10 MPa for 1 h, and flexible samples with a thickness of ~0.4 mm were obtained as a result.

The contents of ceramic and carbon fillers in the polymer matrix were selected taking into account our previous studies [31,34]. The introduction of both fillers in concentrations up to the percolation threshold is more effective, as it increases the permittivity. The addition of fillers in an amount above the percolation threshold practically does not increase the permittivity, while such concentrations significantly increase the dielectric losses.

## 3. Results and Discussion

The synthesized ceramic powder was studied by the XRF method to confirm its chemical composition and structure. The XRF spectrum of KFTO(H) powder was used for quantitative elemental analysis (Figure 2). The XRF spectrum has fluorescence bands, which confirm the presence of K, Fe and Ti in the sample composition. According to the estimated quantitative ratio of the metal oxides and hollandite-like structure, corresponding to the K_x_(Me,Ti)_8_O_16_ stoichiometric formula, the obtained hollandite could be characterized as K_1.8_Fe_1.8_Ti_6.2_O_16_ solid solution, which is similar to the theoretical formula used for the synthesis (K_1.6_Fe_1.6_Ti_6.4_O_16_).

X-ray diffraction patterns of the original KFTO(H) powder and KFTO(H)/PVDF composites with various amounts of KFTO(H) filler, including pure PVDF obtained by polymerization from a DMF solution followed by hot pressing, are shown in Figure 3. It is known that PVDF is a semi-crystalline polymer that can exhibit various crystalline phases, which can be found alone or as mixtures in an amorphous molecular environment, and the dielectric properties can be determined. In our case, diffraction reflections of the α-, β- and γ-phases were found in the diffraction patterns. The reflections at 2ϴ angles equal to 17.8 and 26.8° were attributed to the presence of the non-polar α-phase, corresponding to the (100) and (021) planes, respectively. The reflections at 2ϴ angles = 20.1, 33.2, 36.1 and 39.1° correspond to the reflections of the polar β-phase with a large dipole moment obtained from the (110), (130), (200) and (131) planes, respectively. The reflection at 2ϴ = 18.6° corresponds to the (020) plane of the polar γ-phase with a smaller dipole moment. The phase composition of PVDF strongly depends on the processing methods, and the type and amount of filler. However, for all the composites, the reflections of the desired polar β-phase are traced. For the initial KFTO(H) powder, narrow high-intensity diffraction reflections only are observed; these reflections correspond to the potassium titanate with the hollandite-like structure related to the space group I4/m. The calculated unit cell parameters of this compound, according to the work [34], are: a = 1.01576 nm and c = 0.29676 nm.

The FTIR spectra reported in Figure 4 and corresponding to pure PVDF and KFTO(H)/PVDF composites with different KFTO(H) volume fractions confirm the mixture of PVDF polymorphs observed in the XRD patterns. The characteristic absorption bands with the most pronounced intensities at 615, 763 cm^−1^ confirm the presence of the non-polar α-phase, while the peaks at 840, 1273 cm^−1^ characterize the ferroelectric β-phase. At the same time, the bands at 883, 1073 and 1410 cm^−1^ have similar characteristics for α-, β- and γ-phases of PVDF [35,36]. The FTIR spectrum of KFTO(H) shows intense absorption bands only at 570 and 805 cm^−1^, which characterize the stretching vibrations of the Ti-O-Ti bonds. For the composites containing KFTO(H), intense peaks of the α- and β-phases are also observed in the FTIR spectra; therefore, it can be assumed that the KFTO(H) filler does not contribute to a decrease in the desired polar β-phase in the composites.

The SEM microphotographs of the KFTO(H) ceramic filler, MWCNTs and their mixture obtained by mechanical mixing using a vibrating mill are shown in Figure 5. It is seen that KFTO(H) powder consists of particles less than 1 µm in size, and these particles have plane facets, typical for the tetragonal hollandite-like structures, and form large agglomerates. The MWCNT carbon filler is represented by fiber-shaped nanoparticles, which can also form agglomerates of 10 µm and more in size. The mechanical mixing of KFTO(H) and MWCNTs leads to separation of large agglomerates of both components into single particles and distribution of MWCNTs among the KFTO(H) particles. So, the use of two-phase filler of KFTO(H)/MWCNTs decreases self-agglomeration of individual particles. It allows expecting an improvement in the dispersion of the fillers in the polymer matrix.

The SEM microphotographs of the KFTO(H)/MWCNT/PVDF composite containing maximal volume fraction of the filler (30 vol.%) are shown in Figure 6. Large KFTO(H) particles are inserted into the PVDF polymer matrix. At the same time, ultrathin MWCNT nanoparticles are embedded in the PVDF matrix independently from KFTO(H) particles. The presence of large agglomerates of the filler was not recognized in the SEM images. That is why the KFTO(H) particles have a good compatibility with the PVDF matrix, and the joint use of the KFTO(H) and MWCNT fillers ensures their uniform distribution in the polymer matrix. An improved filler-to-matrix connection can contribute polarization processes in the interface.

Frequency dependences of permittivity, dielectric loss tangent, conductivity and Cole–Cole plot (ε′–ε″) for the composites obtained with different volume fractions of the ceramic filler KFTO(H) measured at room temperature are shown in Figure 7. It is seen that the permittivity of pure polymer matrix PVDF and its dielectric losses are 11 and 0.016 at *f* = 10 kHz, respectively. It indicates the predominance of the polarizable β-phase. The insertion of the KFTO(H) particles into the PVDF matrix decreases the value of permittivity to ε′ = 10, but dielectric loss tangent increases up to 0.022. It may be primarily associated with a decrease in the content of the β-phase in the PVDF composition. A following increase in the KFTO(H) content promotes an increase in permittivity and dielectric losses. However, at maximal filler concentration (30 vol.%), we see a strong increase in permittivity and dielectric losses. This effect can be explained taking into account that as frequency increases the space charge polarization disappears and dipole polarization plays a dominant role. Such behavior of a frequency dependence of the dielectric properties at high concentration of the ceramic filler takes place, presumably, due to reduced dispersibility of this filler in the PVDF matrix.

A value of the dielectric constant for the KFTO(H)/MWCNT/PVDF composites containing 0.5 wt.% of MWCNTs is frequency-independent regardless of the contents of the ceramic filler (Figure 8a). Permittivity values increase with an increase in the filler volume fraction up to 20–22 at 30 vol.%, whereas dielectric losses significantly decrease in the presence of the MWCNTs down to ~0.13 at 30 vol.% (Figure 8b). The value of conductivity has linear frequency dependence (Figure 8c) and the Cole-Cole plot consists of two semicircles (Figure 8d).

The general behavior of the frequency dependencies for permittivity, dielectric losses, conductivity and Cole–Cole plot for the composites containing up to 1.0 wt.% of MWCNTs is similar. It is only necessary to note a significant increase in the corresponding characteristics for the composites with 30 vol.% of RFTO(H) (Figure 9).

For the three-phase polymer matrix composites containing the maximal addition of MWCNTs, all the studied dielectric characteristics somewhat decreased, however, the trend of the corresponding curves is similar to the previous ones (Figure 10).

As shown in Figure 7, Figure 8, Figure 9 and Figure 10, an increase in the MWCNT concentration reduces the contribution of the space charge polarization with a decrease in the frequency of the external electric field. As a result, the frequency dependence of permittivity becomes smoother, and the value of the dielectric loss tangent decreases to the acceptable level of 0.05–0.08 at 1–10 kHz. As mentioned above, the introduction of MWCNT particles into the ceramic filler promotes more efficient separation of KFTO(H) agglomerates and improves its dispersion in the polymer matrix. As seen from Figure 7, Figure 8, Figure 9 and Figure 10, the addition of the conductive MWCNT filler improves the compatibility between the PVDF matrix and KFTO(H) filler phase, which increases the effective permittivity of the composites. In addition, the formation of microcapacitors at MWCNT conductive particles in the polymer matrix promotes increased permittivity of the composites. However, simultaneously with an increase in permittivity, increased contents of MWCNTs encourages an increase in dielectric losses. This phenomenon arises due to an increase in the role of interfacial polarization, current conduction and/or a movement of molecular dipoles.

An increase in the volume fraction of the filler, as well as an increase in the concentration of MWCNT conductive particles, contributes to an increase in dielectric losses. This effect takes place, primarily, due to the fact that an increase in the amount of KFTO(H) leads to an increase in the interfacial area and the formation of small domains inside the polymer matrix. In addition, according to the percolation theory, an increase in MWCNT contents promotes the formation of conductive channels and, consequently, an increase in dielectric losses. This effect has been further confirmed by electrical conductivity analysis. It is seen that an increase in MWCNTs leads to an increase in the conductivity of composites from 9.6 × 10^−8^ S∙cm^−1^ (without MWCNTs) up to 1.9 × 10^−7^ S∙cm^−1^ (1.5 wt.% MWCNTs) at the filler volume fraction of 30 vol.% at *f* = 10 kHz. In Figure 7d, Figure 8d, Figure 9d and Figure 10d, the ε′–ε″ plot shows different relaxation characteristics for the KFTO(H)/MWCNT/PVDF composites. The ε′–ε″ plots of the composites consist of two arcuate curves, and the diameters of these arcs increase with increased contents of the filler. These arcs arise from two polarization mechanisms, namely dipole relaxation at high frequencies and interfacial polarization at lower frequencies due to the presence of inorganic fillers. For all the composites, it should be noted that there is an increase in the proportion of the low-frequency arc with increasing the filler concentration. This fact confirms the increase in the proportion of the interfacial polarization in highly filled composites.

As shown in Figure 11a, the use of 0.5 wt.% MWCNT additive in the filler composition slightly reduces the value of permittivity relative to the composites containing pure KFTO(H). An increase in the content of MWCNTs to 1 wt.%, provides a significant increase in the permittivity, observed relative to the composites with pure KFTO(H) and KFTO/MWCNTs (0.5 wt.%). In the case of the filler content of 15 vol.%, the permittivity values are 17, 17 and 26 at MWCNT content in the filler of 0, 0.5 and 1 wt.%, respectively. An increase in the content of nanotubes up to 1.5 wt.% leads to a decrease in the permittivity relative to composites containing 1 wt.% MWCNTs; however, the value of permittivity is higher than those for the composites containing 0.5 wt.% MWCNTs and pure KFTO(H). In the case of a filler volume content of 15 vol.%, the value of permittivity decreases from 26 to 22 with an increase in the concentration of MWCNTs from 1 to 1.5 wt.%. This behavior can be explained by the theory of the presence of two types of microcapacitors in three-phase composites: (1) capacitors with plates of MWCNTs separated by the ceramic dielectric; (2) capacitors with plates of MWCNTs separated by the polymer dielectric. For the composites with 0.5 wt.% of MWCNTs, the appearance of the second type of capacitors somewhat reduces the permittivity, since microcapacitors of the second type have low values of capacitance. A decrease in the dielectric constant values with increased content of MWCNTs, from 1 to 1.5 wt.%, can be explained by the shunting of some of the capacitors of the first type due to increased numbers of the conductive nanotubes.

The balance between permittivity and dielectric loss tangent values is important for the polymer matrix composite dielectrics. In a recently published paper [43], an approach is proposed for dividing the *ε*′–*tanδ* diagram by a line described with the equation ε = 76.6 lg(*tanδ*) + 153.2 (Figure 11b) based on the analysis of numerous reviews. The area up to this line (non-colored) has values of dielectric parameters, which are satisfactory for use in electronic devices. To analyze the balance of *ε*′–*tanδ* for the obtained dielectrics, our data for the composites containing 30 vol.% of the filler and characterized by the highest permittivity are represented in Figure 11b together with some literature data; the arrows indicate an increase in the MWCNT fraction. This diagram illustrates that despite a decrease in permittivity with the addition of 0.5 wt.% MWCNTs, the characteristics of this composite approach the indicated line due to a decrease in dielectric losses. A further increase in MWCNT content up to 1 wt.% also increases the permittivity and, despite an increase in the dielectric loss tangent, brings the composite parameters into the region of optimal balance. A further increase in the MWCNT content leads to a decrease in permittivity and makes the values of *ε*′ and *tanδ* less balanced relative to this series. The resulting composites can be used as dielectrics for passive elements of flexible electronics.

## 4. Conclusions

Polymer composites based on PVDF matrix with ceramic filler (5.0, 7.5, 15 and 30 vol.%) and MWCNT admixtures (0.5, 1.0 and 1.5 wt.% of the filler) were produced and investigated. The hollandite-like ceramics named as KFTO(H) and characterized by the chemical formula of K_1.8_Fe_1.8_Ti_6.2_O_16_, determined using XRD and XRF analyses, were applied as a filler.

The characteristic absorption bands of α-, β- and γ-phases of the polymer as well as the ceramic filler were recognized in the FTIR spectra of the composites. It has been established that neither the destruction processes nor any other chemical interactions happened during the composite production. Different ratios of the PVDF and KFTO(H) were confirmed by intensities of the reflections (XRD patterns) and absorption bands (FTIR spectra). The SEM images indicated that the MWCNT additives provided better dispersibility of the ceramic filler in the polymer matrix and favored improved dielectric properties.

It has been shown for the first time that the use of a finely dispersed ceramic filler, which is a hollandite-like solid solution, in combination with a conductive additive in the form of MWCNTs, makes it possible to increase the dielectric constant and simultaneously reduce the dielectric losses of composites with a polymer matrix based on PVDF. The optimal balance of permittivity and dielectric losses was found for the composite containing 30 vol.% of KFTO(H) and 1.0 wt.% of the MWCNT additive (ε′ = 77.6 and *tan(δ)* = 0.104 at 1 kHz).

The obtained results showed the promise of using the investigated composites as dielectrics in the production of passive elements of flexible electronics.

## Figures and Tables

**Figure 1 polymers-14-04609-f001:**
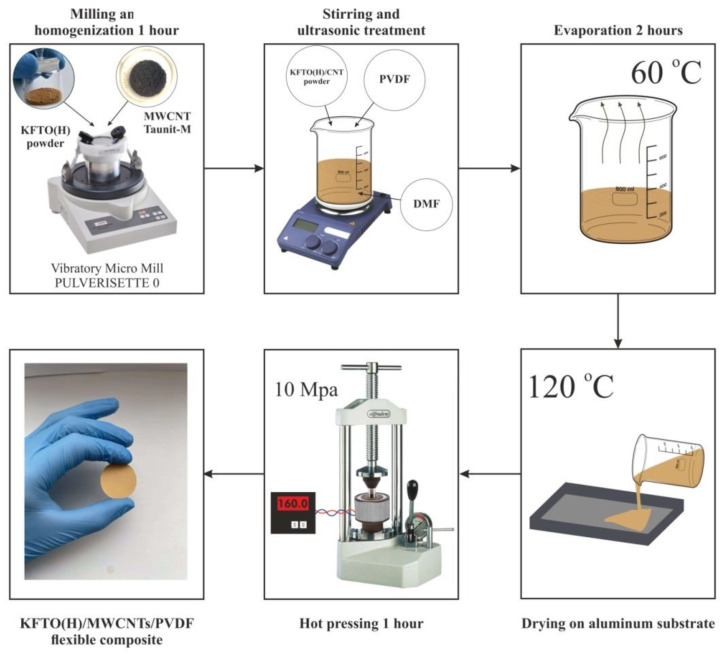
Composite preparation methodology.

**Figure 2 polymers-14-04609-f002:**
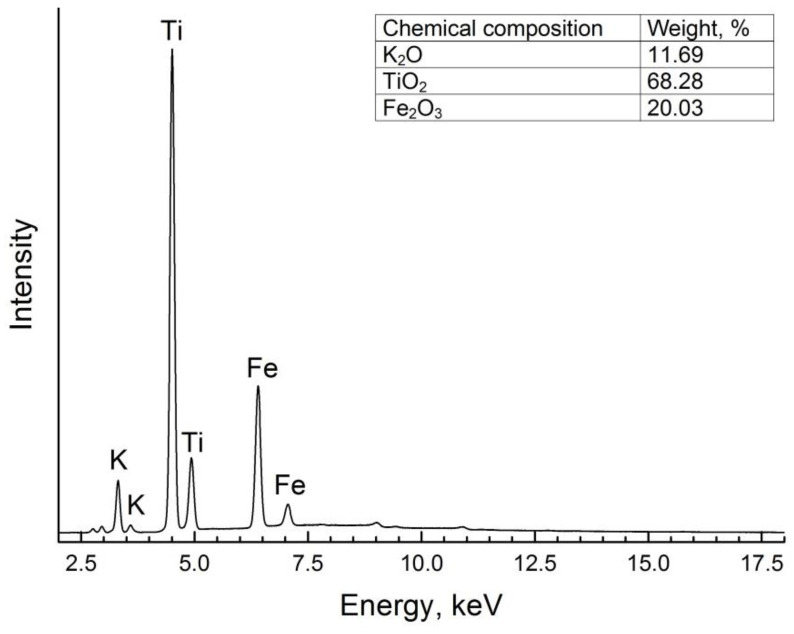
XRF spectrum of KFTO(H) powder.

**Figure 3 polymers-14-04609-f003:**
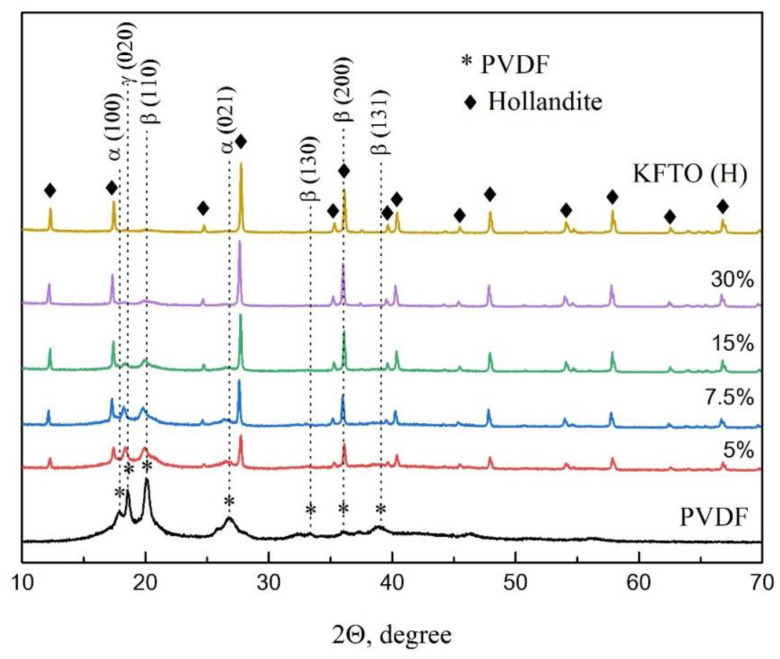
XRD patterns of the KFTO(H)/PVDF composites.

**Figure 4 polymers-14-04609-f004:**
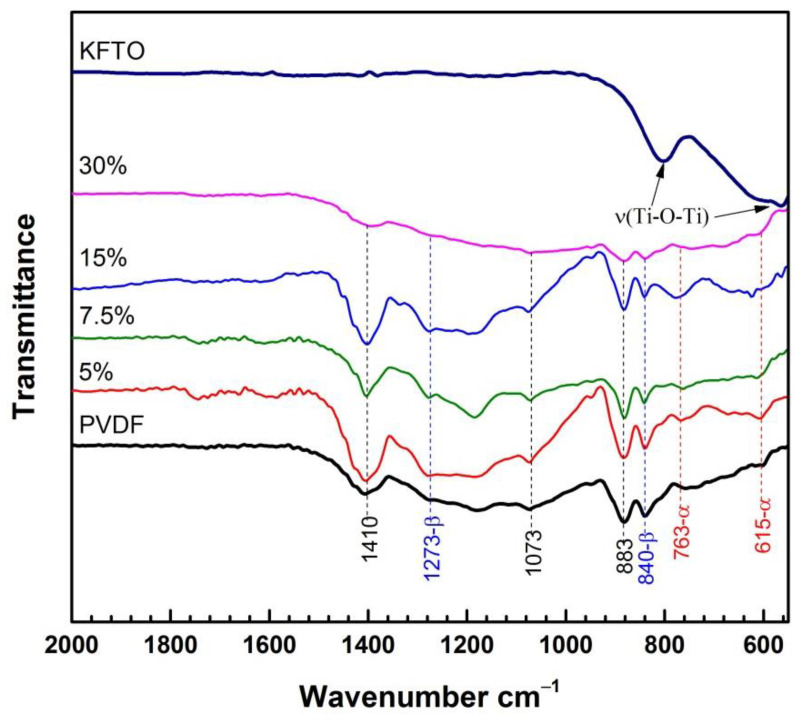
FTIR of the KFTO(H)/PVDF composites.

**Figure 5 polymers-14-04609-f005:**
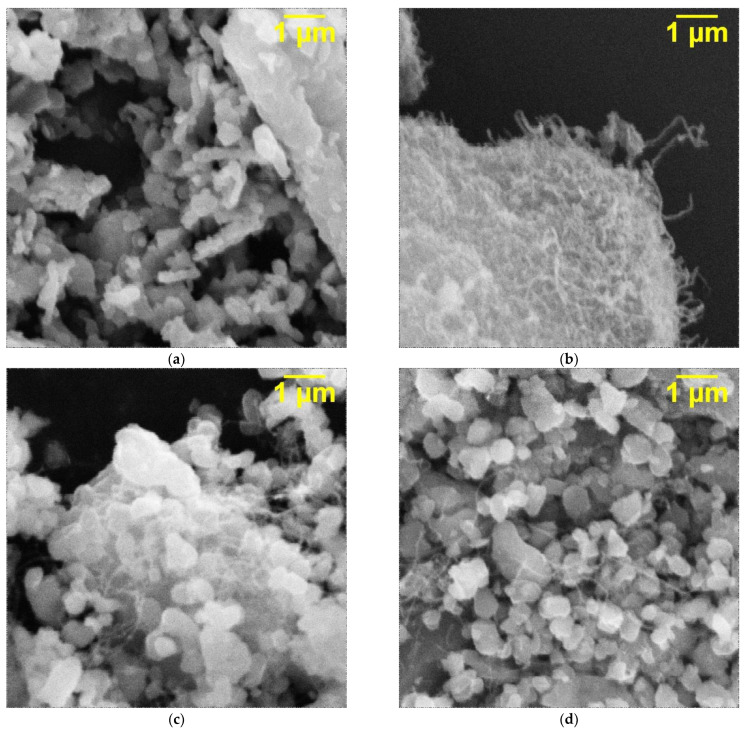
SEM microphotographs of (**a**) KFTO(H) powder, (**b**) MWCNTs, (**c**,**d**) composite KFTO(H)/MWCNTs (MWCNTs 1.5 wt.%).

**Figure 6 polymers-14-04609-f006:**
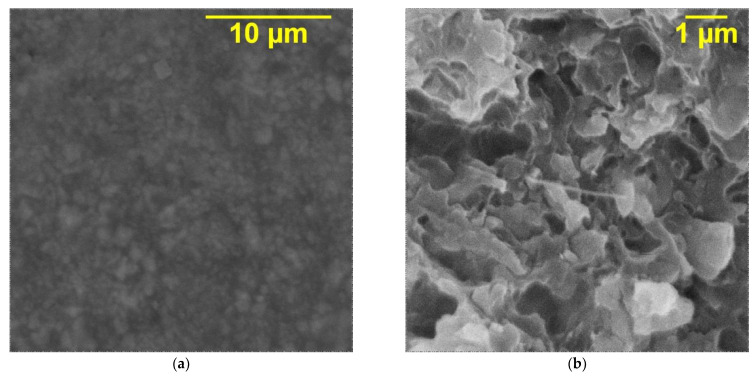
SEM microphotographs of (**a**) surface and (**b**) cross-section of three-phase composites of KFTO(H)/MWCNTs/PVDF with 30 vol.% of filler.

**Figure 7 polymers-14-04609-f007:**
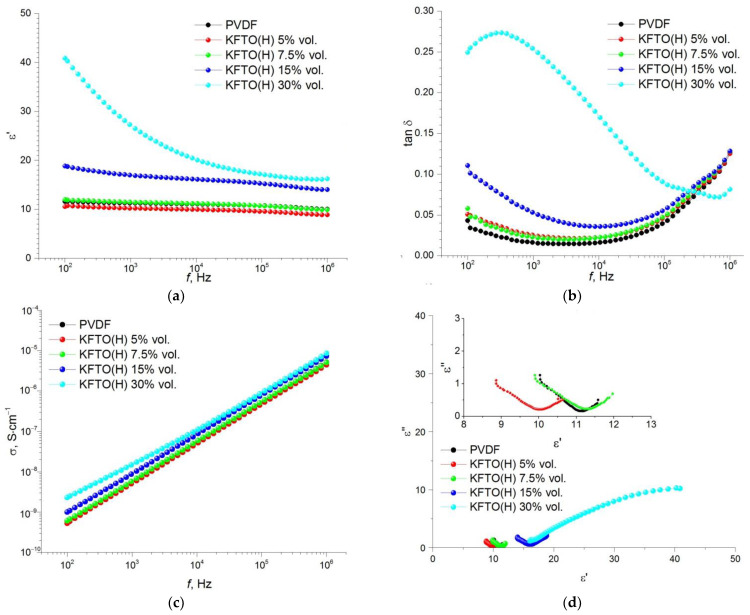
Frequency dependence of (**a**) permittivity, (**b**) dielectric loss tangent, (**c**) conductivity and (**d**) Cole–Cole plot (ε′–ε″) for PVDF-based composites with different volume fractions of KFTO(H) (inset is the enlarged view of the basic figure).

**Figure 8 polymers-14-04609-f008:**
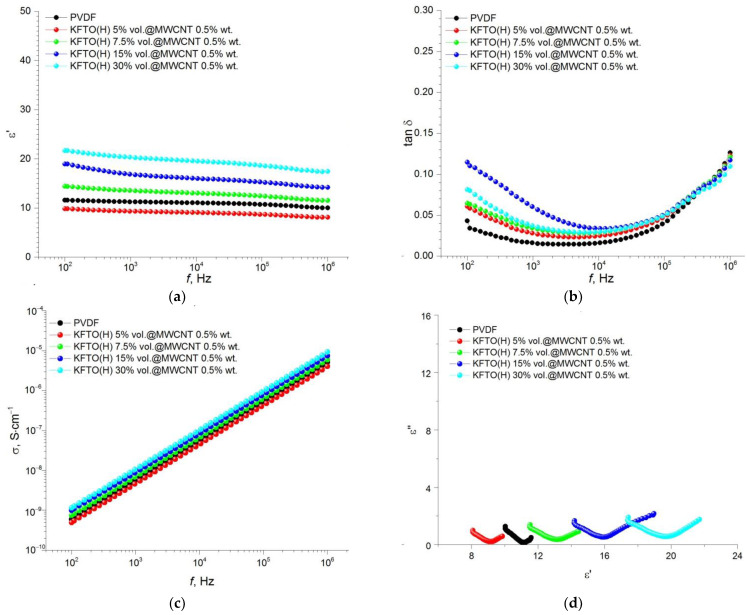
Frequency dependence of (**a**) permittivity, (**b**) dielectric loss tangent, (**c**) conductivity and (**d**) Cole–Cole plot (ε′–ε″) for PVDF-based composites with different volume fractions of KFTO(H) with MWCNT additive of 0.5 wt.%.

**Figure 9 polymers-14-04609-f009:**
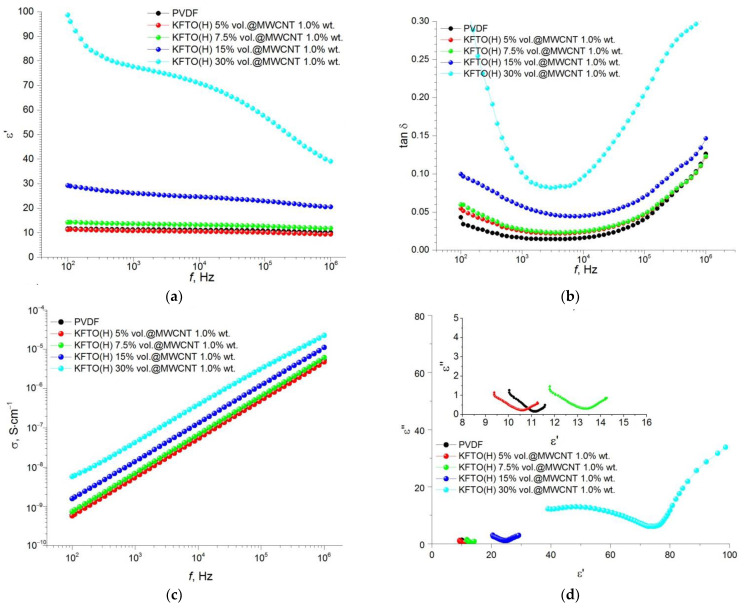
Frequency dependence of (**a**) permittivity, (**b**) dielectric loss tangent, (**c**) conductivity and (**d**) Cole–Cole plot (ε′–ε″) for PVDF-based composites with different volume fractions of KFTO(H) with MWCNT additive of 1.0 wt.% (inset is the enlarged view of the basic figure).

**Figure 10 polymers-14-04609-f010:**
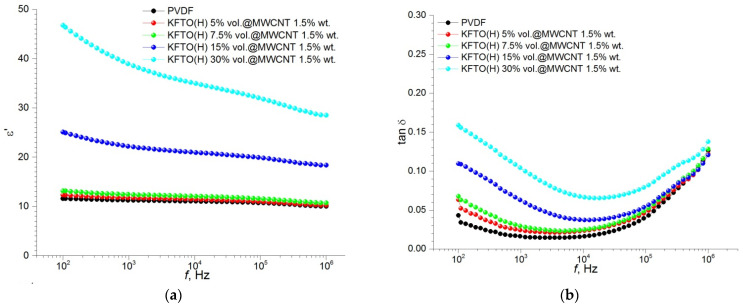
Frequency dependence of (**a**) permittivity, (**b**) dielectric loss tangent, (**c**) conductivity and (**d**) Cole–Cole plot (ε′–ε″) for PVDF-based composites with different volume fractions of KFTO(H) with MWCNT additive of 1.5 wt.% (inset is the enlarged view of the basic figure).

**Figure 11 polymers-14-04609-f011:**
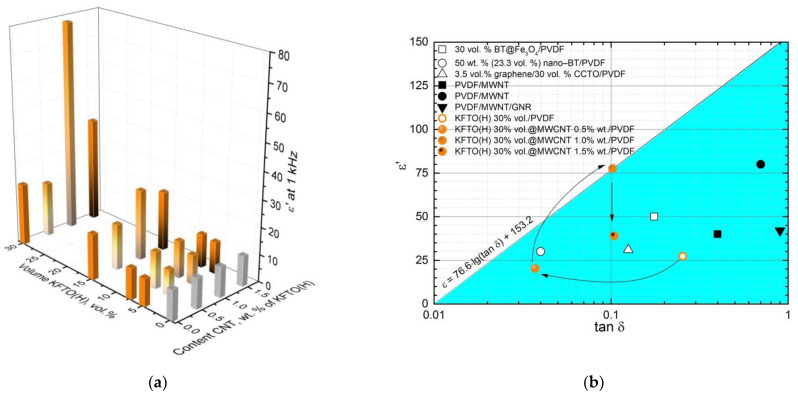
Volume of KFTO(H) and MWCNT content dependence of composite permittivity (gray columns—unfilled PVDF, orange columns without gradient—PVDF filled with KFTO(H), orange columns with gradient—PVDF filled with KFTO(H) and MWCNT). (**a**) and ε′ and *tanδ* values of each sample collected from 1 kHz (the blue background is an area of low balance for ε′ and *tanδ*) (**b**) [37,38,39,40,41,42].

## Data Availability

Not applicable.

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
