# Peer review of "Permittivity and Dielectric Loss Balance of PVDF/K1.6Fe1.6Ti6.4O16/MWCNT Three-Phase Composites"

_polymers, 2022, doi:10.3390/polym14214609_

Round 1

Reviewer 1 Report

In this work, the authors have produced ceramic materials and studied by XRD, FTIR, SEM and etc. Based on the abundant data, the authors found an optimal combination of permittivity and dielectric losses for a certain material. These results are important for future application. Therefore, the reviewer believes that the manuscript can be accepted after a minor revision.

1. Figure 5, what is the difference between c, and d? It seems that the two sub-figures are referring to the same material but looks different.

2. The novelty of this work should be further discussed. After I read through the manuscript, I find that the data are quite clear, but the novelty and its application are not fully discussed. It would nice if the authors discuss these in the abstract and conclusion, instead of only reporting that “xxx has the optimal property”.

3. Figure 7d, 9d, 10d, please indicate that the figure inset is the enlarged view of the big figure. Meanwhile, there is a wired black line below Figure 10d and should be cleaned.

4. Please briefly indicate why choose 30 vol% for KFTO(H) and 1.5 wt% for MWCNT. In the other words, why not e,g, 50% or 10 wt% or higher? There should be a special reason for the choice.

Author Response

Point 1: Figure 5, what is the difference between c, and d? It seems that the two sub-figures are referring to the same material but looks different.

Response 1: Thank you for your comment. One micrograph shows agglomerated particles of material, the other shows non-agglomerated particles of the same material.

Point 2: The novelty of this work should be further discussed. After I read through the manuscript, I find that the data are quite clear, but the novelty and its application are not fully discussed. It would nice if the authors discuss these in the abstract and conclusion, instead of only reporting that “xxx has the optimal property”.

Response 2: Thank you for your comment. We added more information about the novelty in Abstract and Conclusion. (in red)

Point 3: Figure 7d, 9d, 10d, please indicate that the figure inset is the enlarged view of the big figure. Meanwhile, there is a wired black line below Figure 10d and should be cleaned.

Response 3: Thank you for your comment. We corrected it. (in red)

Point 4: Please briefly indicate why choose 30 vol% for KFTO(H) and 1.5 wt% for MWCNT. In the other words, why not e,g, 50% or 10 wt% or higher? There should be a special reason for the choice.

Response 4: Thank you for your comment. According to previous studies, the introduction of both ceramic and conductive filler in concentrations up to the percolation threshold is more effective, as it increases the permittivity. The addition of fillers in an amount above the percolation threshold practically does not increase the permeability, while this significantly increases the dielectric loss. Therefore, we investigated the indicated concentrations. We added this explanation to the methodology for composites producing. (in red)

Reviewer 2 Report

the manuscript is interesting and meets the purposes of the journal but the introduction should show more the purpose and interest of the research

Authors should improve their introduction by taking into account the following questionsWhat is the main question addressed by the research?

What does it add to the subject area compared with other published material?

Author Response

Point 1: Authors should improve their introduction by taking into account the following questionsWhat is the main question addressed by the research?

Response 1: Thank you for your comment. We added more information in Introduction. (in red)

Point 2: What does it add to the subject area compared with other published material?

Response 2: Thank you for your comment. We added more discussion. (in red)

Reviewer 3 Report

The manuscript " Permittivity and dielectric losses balance of PVDF/K1.6 Fe1.6Ti6.4O16/MWCNTs three -phase composites" is an interesting work that contains a lot of work and many data. However, the VERY LOW QUALITY of the English language strongly hinders the text reading and understanding. Besides of this, I make the following comments:
- Lines 88-90 "Optimal stoichiometric ratios of Ti-containing precursor with crosslinking agent and metal ion chelating agent as well as acid residue were experimentally found as Ti:С6H8O7=1.5; Ti:C2H6O2=5.5; Ti:NO3=1.8, respectively". How? An explanation is required

- Line 199-200 " All this together can improve the dielectric properties". Further explanations are necessary.

- The authors do not give enough insight into the discussion on the results of Figures 7-11. This must be significantly improved.

-

Author Response

The manuscript " Permittivity and dielectric losses balance of PVDF/K1.6 Fe1.6Ti6.4O16/MWCNTs three -phase composites" is an interesting work that contains a lot of work and many data. However, the VERY LOW QUALITY of the English language strongly hinders the text reading and understanding. Besides of this, I make the following comments.

The quality of the English language has been improved.

Point 1: Lines 88-90 "Optimal stoichiometric ratios of Ti-containing precursor with crosslinking agent and metal ion chelating agent as well as acid residue were experimentally found as Ti:С6H8O7=1.5; Ti:C2H6O2=5.5; Ti:NO3=1.8, respectively". How? An explanation is required

Response 1: Thank you for your comment. We added more information in KFTO(H) synthesis. (in red)

Point 2: Line 199-200 " All this together can improve the dielectric properties". Further explanations are necessary.

Response 2: Thank you for your comment. This sentence was deleted. Discussion about dielectric properties and factors, influencing them, is in description of Figures 7-11. (in red)

Point 3: The authors do not give enough insight into the discussion on the results of Figures 7-11. This must be significantly improved.

Response 3: Thank you for your comment. The description was added and improved. (in red)

Round 2

Reviewer 2 Report

the authors have an important contribution in the preparation of dielectric materials, they have made modifications to their manuscript enriching their presentation, highlighting the important aspects of their research for which they accept the manuscript for publication